# Vitamin D Supplementation and Its Impact on Different Types of Bone Fractures

**DOI:** 10.3390/nu15010103

**Published:** 2022-12-25

**Authors:** Jakub Erdmann, Michał Wiciński, Paweł Szyperski, Sandra Gajewska, Jakub Ohla, Maciej Słupski

**Affiliations:** 1Department of Pharmacology and Therapeutics, Faculty of Medicine, Collegium Medicum in Bydgoszcz, Nicolaus Copernicus University, M. Curie 9, 85-090 Bydgoszcz, Poland; 2Department of Plastic, Reconstructive and Aesthetic Surgery, Faculty of Medicine, Collegium Medicum in Bydgoszcz, Nicolaus Copernicus University, M. Curie 9, 85-090 Bydgoszcz, Poland; 3Department of Hepatobiliary and General Surgery, Faculty of Medicine, Collegium Medicum in Bydgoszcz, Nicolaus Copernicus University, M. Curie 9, 85-090 Bydgoszcz, Poland

**Keywords:** vitamin D, bone fracture, vertebral fracture, hip fracture, stress fracture, pediatric fracture

## Abstract

Vitamin D helps to balance the levels of calcium and phosphorus to maintain proper bone structure. It is also involved in essential biological roles and displays a wide spectrum of potential benefits in the human body. Since there are many types of fractures that occur at specific ages and due to different circumstances, the influence of vitamin D on the frequency of a particular fracture may differ. Thus, the authors investigated the possible preventive effect of vitamin D on the risks of vertebral fractures, hip fractures, stress fractures and pediatric fractures. Additional aspects of vitamin D, especially on recuperation after injures and its impact on the severity of particular fractures, were also discussed. It was suggested that vitamin D supplementation may contribute to a reduction in hip fracture risk due to reduced bone turnover, decreased frequency of falls and improved muscle function. Furthermore, vitamin D appears to lower the risk of stress fractures in athletes and military recruits. Due to a nonunified protocol design, presented investigations show inconsistencies between vitamin D supplementation and a decreased risk of vertebral fractures. However, a vitamin D preventive effect on pediatric fractures seems to be implausible.

## 1. Introduction

Vitamin D3 is a secosteroid produced endogenously in response to the skin’s exposure to sunlight. The ultraviolet radiation type B (UVB; wavelengths of 290 to 315 nm) is absorbed by 7-dehydrocholesterol in the lower layers of the epidermis and converted to vitamin D3 (cholecalciferol) [1]. Before reaching the active form, vitamin D3 is hydroxylated to 25-hydroxyvitamin D (calcidiol) in the liver and, afterwards, to its final form, 1,25-dihydroxyvitamin D (calcitriol), in the kidneys [2]. It is estimated that UVB-induced skin synthesis accounts for 90% of vitamin D3 production in the human body [3]. However, the mentioned synthesis can be inhibited by the higher presence of melanin, which absorbs ultraviolet radiation and explains the increased risk of vitamin D deficiency in African Americans [4]. What is more, UVB absorption can be impaired by sunscreen use, season, altitude, latitude, time of the day and even air pollution [3,5]. Although there is a natural synthesis of vitamin D3 in the human body, it still must be delivered from external sources such as a dietary supplement, drugs or food for maintaining its optimal status [1]. The highest content of vitamin D3 is found in animal products such as fishes, offal, egg yolks and butter [6]. With the exception of mushrooms, which are rich in ergosterol (a precursor of vitamin D2 (ergocalciferol)), edible plants are not considered as an additional source of vitamin D3 [7]. However, due to the fungal contamination of cocoa beans, a high concentration of vitamin D2 was found in cocoa powder and cocoa butter [8]. Vitamin D2 is synthesized from ergosterol after exposure to ultraviolet irradiation in mushrooms. The differences in the structures between vitamin D2 and D3 are that vitamin D2 has a side chain, and it contains a double bond between carbon-22 and carbon-23 and a methyl group on carbon-24 [9]. Although there are the differences in their structures, the same enzymes in the human body convert them into active compound calcitriol. However, their efficacy in raising the vitamin D status in humans is not equal, and vitamin D3 should be the preferred choice for supplementation [10,11,12]. The inequality in effectiveness may be explained by a lower degree of attachment of vitamin D2 to the vitamin D binding protein or its impaired hydroxylation at C25 [13]. In addition to the mentioned factors that inhibit vitamin D3 synthesis, there are unmodifiable factors such as genetic determinants, ethnicity and sex that impact the vitamin D3 status. Female gender is associated with lower vitamin D3 levels and more severe deficiency compared to males [14]. This distinction may result from the excess body fat tissue of women that decreases the circulating level of vitamin D3 [15]. Among the modifiable risk factors that are linked to the lower plasma vitamin D3 level, there can be distinguished obesity, smoking cigarettes and poor eating habits [3,16].

The active form of vitamin D (calcitriol) mediates the biological effects by binding to the vitamin D receptor (VDR), which is located in the nuclei of target cells [17]. VDRs are expressed mainly by pancreatic islets, renal distal tubules, intestinal enterocytes and osteoblasts [18]. The calcitriol–VDR connection allows to start the cascade of processes that modulates the gene expression of specific proteins, which play different roles in a variety of biological actions, including the maintenance of the phosphorus and calcium levels in the blood [19]. Vitamin D stimulates calcium and phosphorus absorption, mostly from the intestines. It is also crucial in the renal absorption of calcium back to the blood vessels and the regulator of the calcium homeostasis in bones [20]. The parathyroid hormones stimulate the release of calcium from bones into the bloodstream. Thus, the elevated level of parathyroid hormones leads to excessive bone turnover, bone loss and mineralization defects [21]. At the level of the parathyroid gland, calcitriol decreases parathyroid hormone synthesis and secretion. It also stimulates calcium sensor receptor (CaR) expression, which causes inhibition of the parathyroid gland by calcium [22]. In the result of the mentioned functions of vitamin D3, calcium and phosphorus are maintained at the optimal levels to deliver proper bone mineralization and prevent them from rickets and osteomalacia [20,22,23].

The health effect of vitamin D3 on the human body is constantly expanded, and in recent years, it has been reported that vitamin D3 supplementation may be beneficial in athletes and in the management of asthma and cardiovascular conditions [24,25,26]. Furthermore, it was suggested that patients with decreased articular cartilage thickness are more likely to be vitamin D insufficient, and low levels of vitamin D are associated with the development of osteoarthritis [27]. This review is an attempt to describe the present state of knowledge about vitamin D supplementation and its impact on the risk of different fractures, as well as to emphasize the additional effects on recuperation after injuries.

## 2. Fracture Risk Assessment and Vitamin D3 Status Guidelines

In the human body, the peak bone mass develops in the third decade of life [28], whereas bone loss starts around the fourth decade [29]. The natural loss of bone mineral density leads to primary osteoporosis, whereas secondary osteoporosis is caused by many factors, including endocrinopathies, chronic inflammatory conditions, chronic kidney diseases, gastrointestinal diseases, cancer therapy, glucocorticoids treatment, alcohol abuse and others [30]. It is estimated that, after the age of 50 years, 20% of men and 50% of women experience low-energy fractures [31]. The postmenopausal period of women’s lives accelerates bones tissue loss due to estrogens deficiency, which explains the abovementioned difference of fracture occurrences between the sexes [32]. Furthermore, the contribution of vitamin D3 deficiency and age-related decrease in calcium absorption may lead to secondary hyperparathyroidism [21,22]. These factors all weaken bones’ structures and increase the risk of fractures [33].

There were created risk assessment tools that help clinicians to estimate their patients’ fracture risks and implement preventive strategies. The introduction of weight-bearing exercise; home hazard removal; patients education and pharmacological treatment such as bisphosphonates, RANKL inhibitors, teriparatide and anti-sclerostin antibodies exerted a positive influence in fracture rates [30,34]. The best-studied algorithms are: The Fracture Risk Assessment Tool (FRAX), the Garvan Fracture Risk Calculator and the QResearch Database QFracture Score [35]. The FRAX calculator is the most commonly used and was developed by the World Health Organization. It is a multifactor calculator that estimates the 10-year absolute risk of hip, vertebra, forearm or proximal humerus fractures in both sexes above the age of 50 years. Taking into account the different levels of health care financing and individual epidemiology of each country, there are various models of FRAX calculators [35,36]. It is worth mentioning that neither FRAX nor the abovementioned tools include the vitamin D3 status.

According to The National Osteoporosis Society and other Vitamin D Guidelines Summaries, the serum vitamin D3 thresholds are the following: (1) <30 nmol/L (12 ng/mL) is deficient, (2) 30–50 nmol/L (12–20 ng/mL) may be inadequate in some people and (3) >50 nmol/L (>20 ng/mL) is sufficient for almost the whole population [37,38,39]. To maintain proper musculoskeletal health among adults, the intake recommendations for vitamin D3 are between 400 and 2000 IU/day, but it depends on individual health status, age, body weight, latitude of residence and diet [37,38,39,40].

## 3. Vertebral Fractures

Vertebral fractures (VF) are the most common osteoporotic fractures, and their prevalence increases by age. In the group over 70 years old, 19% of women and 20% of men are affected by VF, and the most frequent type of fracture is the wedge type [41]. Only one-fourth to one-third of VFs are detected clinically, not just radiologically [42]. They reduce the health-related quality of life and represent the major cause of morbidity among postmenopausal women [43]. Moreover, clinical VFs are associated with an increased risk of mortality [44], even in a group of relatively healthy older women [45]. As VFs indicate an increased risk of future fractures, it has been suggested that patients with VFs should be evaluated by a multidisciplinary team for the underlying and reversible medical conditions of secondary osteoporosis [46]. Hyperthyroidism, uncontrolled hypothyroidism, hypercalciuria, low testosterone levels, and vitamin D deficiency were distinguished as potentially correctable contributors [47]. Thus, early prevention of that type of fracture seems to be a reasonable strategy, especially if vitamin D and calcium supplementation are estimated as a low-cost intervention [48]. Lopes et al., in their prospective study, presented that the serum level of vitamin D below 75 nmol/L (30 ng/mL) was a significant factor for VFs, the same as age and femoral neck body mineral density among elderly women in Brazil [49]. Similar conclusions were stated by Nakamura et al., who found out in their cohort study that the serum level of vitamin D above 71 nmol/L (28.5 ng/mL) was associated with a reduced risk of VF in Japanese older women [50]. The protective effect of vitamin D3 was also suggested by Maier et al., who provided statistically significant differences in the serum level of vitamin D3 between a VF group and control group with back pain (mean serum level of vitamin D 49.1 nmol/L (19.67 ng/mL) versus 62.6 nmol/L (25.08 ng/mL), respectively). What is more, a difference between vitamin D and gender was not observed [51]. Likewise, Zhang et al. evaluated the serum level of vitamin D between 534 patients with VFs and 569 orthopedic patients with back pain in China. The authors found that, if the mean serum vitamin D concentration of participants was 29.67 ± 6.18 nmol/L (11.89 ± 2.48 ng/mL), the risk of VF increased almost twofold, and subjects suffered more severe fractures. Furthermore, when the mean serum concentration was in a range from 60.91 nmol/L (24.40 ng/mL) to 103.3 nmol/L (41.39 ng/mL), the VF risk was significantly lower [52]. Interestingly, a preventive effect of vitamin D3 was also confirmed among different medical conditions. The study of Hernández et al. presented that vitamin D3 concentrations above 63.6 nmol/L (25.5 ng/mL) decreased bone turnover stimulated by high levels of parathormone (>58 pg/mL) and reduced the VF prevalence in normocalcemic postmenopausal women with high parathyroid hormone levels [53]. Moreover, the protective effect of a higher serum level of vitamin D against VF was also observed in patients on hemodialysis [54] and in patients after kyphoplasty [55]. On the contrary, the research performed by Pramyothin et al. showed no association between the VF and vitamin D3 levels in 495 postmenopausal Japanese women living in Hawaii [56]. However, the mean serum level of vitamin D3 among this population was 101.4 nmol/L (40.6 ng/mL), which may be considered relatively high, and only 44 (8.9%) subjects constituted a level below 63.6 nmol/L (25.5 ng/mL). This low prevalence of vitamin D insufficiency may explain the lack of association in this investigation. The mentioned studies allow the assumption that maintaining vitamin D3 baseline levels above 60–70 nmol/L (24–28 ng/mL) is enough to trigger a protective effect against VF in older adults.

As the results of observational studies appear to be consistent, experimental research has not provided a similar conclusion. In 2002, Papadimitropoulos et al. demonstrated in their meta-analysis of eight randomized clinical trials with 1130 postmenopausal women that hydroxylated vitamin D, mostly in combination with calcium, reduced the incidence of vertebral fractures [57]. However, in 2007, the meta-analysis of Jackson et al., which included a similar number of cases and the patient health profiles, did not confirm that result [58]. Subsequent meta-analyses, which were based on a higher number of patients of different sexes, compared the preventive effects of vitamin D alone, calcium alone and the combination of vitamin D and calcium versus a placebo, but none of them reduced the risk of VF [59,60]. The discrepancies between studies may arise from several reasons. First of all, only a few clinical trials reported the vitamin D3 baseline level among the subjects and its level after treatment, which does not allow assessing if a potential preventive level of vitamin D3 was achieved. Secondly, the average dose of vitamin D3 was mostly 300–800 IU [58,59], which might be considered as too low to reduce the risk of fractures in the older population. For example, it was found that a daily vitamin D3 intake of 1000 IU increases the vitamin D3 concentration by approximately 12–25 nmol/L (4.8–10 ng/mL) [61]. Another study presented that a vitamin D3 supplementation of 5000 IU for 12 months increased the baseline vitamin D3 level from 59 nmol/to 169 nmol/L (23.6–67.6 ng/mL) [62]. Furthermore, the vitamin D3 dose should be adequate to the patient’s weight, because obese subjects have 8 nmol/L (3.2 ng/mL) lower serum vitamin D3 levels, and they should be administered two or three times higher doses of vitamin D3 [61]. What is more, the included clinical trials in the abovementioned meta-analyses used different time intervals of oral vitamin D supplementation. Some investigations showed that daily vitamin D administration is more effective than weekly, whereas monthly supplementation is the least successful [63]. Summarizing, the vitamin D3 efficacy in preventing VF is inconsistent, and aspects that may explain mismatches between investigations should be taken into account in protocol designs of future studies.

A summary of the studies describing the impacts of vitamin D on VFs is shown in Table 1 and in Table 2.

## 4. Hip Fractures

A hip fracture (HF) is one of the most serious complications of osteoporosis and is associated with high morbidity and mortality [64]. In the medical fields, they are called “the last fracture of the elderly”. Anatomically, there can be distinguished fractures in the area of the femoral neck (intracapsular) and intertrochanteric (noncapsular) fractures. Most of the conducted studies do not consider these types of fracture separately [65]. Apart from death, they may lead to a significant deterioration of the physical condition and loss of independence [66]. Postmenopausal women are the most significant group in HF incidents. Statistically, post-fracture mortality in Europe and the USA ranges from 10% to 15% in women over the age of 50, while the risk for men is one-third or half of these figures. [67]. The frequency of HF appears to increase worldwide, mainly due to the higher life expectancy. It is estimated that the number of HFs will be 2.6 million by the year 2025 and 4.5 million by the year 2050 [67].

Conditions such as impaired functional mobility, neurologic disorders, visual defects, muscular weakness and decreased proprioception are risk factors for falls that account for 95% of HF cases. Furthermore, a prior fragility facture, cigarette smoking, benzodiazepines and psychotropic drugs usage, corticosteroids, functional impairments, conditions affecting bone strength and others are identified as independent risk factors for HFs [68]. It is logical to hypothesize that a vitamin D3 deficiency may be another risk factor for HFs due to its participation in the regulatory processes of calcium and skeletal homeostasis [66,69]. The study of Lauretani et al. conducted on 974 Italian patients with an average age of 85.7 admitted to hospital because of HFs showed that 84.2% of them had a vitamin D3 deficiency [70]. Similar relationships were presented by Johnson et al. [71] and Neale et al. [72], who conducted studies among patients who live in sunny areas.

Another direction of research was based on the relationship between vitamin D3 supplementation and the risk of fractures. Lilliu et al., in their study, showed a reduction in the incidence of HFs in patients who supplemented vitamin D at a dose of 800 IU and 1200 mg of calcium compared to the placebo group. After 36 months of observation, there were 25% fewer cases of HFs in the study group compared to the placebo group [64]. A protective effect of vitamin D3 against HFs was also demonstrated by Rossini et al. They administrated 400,000 IU of vitamin D per year in women aged >65 years and noted a decreased frequency of HFs by 10% compared to the control group [73]. The authors suggested that a combination of vitamin D3 and calcium may reverse secondary hyperparathyroidism, which decreases bone turnover [64,73]. However, some research presented that vitamin D exerts a positive impact on muscle tissue, and myopathy could be a prominent symptom before the biochemical signs of bone deterioration [25,74]. The increased muscle strength may decrease the incidence of falls, which is the main cause of HFs. Bischoff et al., in their study, showed that, after 3 months of vitamin D3 and calcium supplementation, the risk of falls decreased by 49% [74,75].

Another scientist investigated the influence of vitamin D on recuperation after hip injury. Stemmle et al. concluded that patients who were administered doses of 800 IU vitamin D3 in combination with simple exercise in the first year after a HF had improved function of the lower extremities [76]. Other scientists showed that a single dose of vitamin D3 may affect the recovery period after hip fracture surgery. It was found that a single dose of 100,000 IU vitamin D3 decreased the incidence of early postoperative complications [66], whereas a dose of 250,000 IU vitamin D3 reduced pain levels and the frequency of falls [77].

There are also studies that challenge the hypothesis about the beneficial effects of vitamin D on the prevention of HFs. Smith et al. conducted a randomized, double-blinded, controlled study on 9440 people over 75 years of age. Patients were administered intramuscularly once a year with a dose of 300,000 IU of vitamin D2 (ergocalciferol). On the basis of the obtained results, the authors concluded that supplementation with vitamin D2 had no effect on the prevention of osteoporotic injuries, including HFs [78]. The main drawback of the study was the administration of ergocalciferol, which is less efficient compared to cholecalciferol [10,11,12,13]. Glendeninng et al. presented that vitamin D3 is superior at raising the serum vitamin D3 level compared to vitamin D2 in patients after HFs [79]. Another study that did not support the vitamin D3 protective effect was performed by Meyer et al., who examined the population of elderly people living in Norwegian nursing homes for 2 years. The study group received a dose of 400 IU of vitamin D3 in cod liver oil, whereas the control group administered cod liver oil without vitamin D3. Although the intervention group increased their vitamin D3 concentration by 22 nmol/L (8.81 ng/mL), no noticeable difference in the incidence of HF was found between the study groups [80]. The result may be due to the fact that people who previously supplemented vitamin D were admitted to the study, or an excessive dose of vitamin A was delivered with cod oil. It was found that hypervitaminosis A may increase the fracture risk due to the inhibition of osteoblast differentiation and bone mineralization [81].

The abovementioned studies showed that vitamin D3 supplementation may decrease the frequency of HF, possibly due to a reduced bone turnover, muscle strengthening and lower incidence of falls. Consequently, decreased serum vitamin D levels seem to be associated with an increased risk of falls and hip fractures. The above analyzed studies also implied that the problem of deficiency applies to very sunny countries [71,72], which suggests the implementation of preventive supplementation with vitamin D3 in the general population, regardless of the place of residence. Studies have also shown the validity of vitamin D3 supplementation after HF as a potential factor improving recuperation [66,76,77]. In summary, most of the conducted studies allowed the conclusion that vitamin D supplementation has a beneficial effect on the prevention of HFs, as well as better convalescence.

A summary of the studies describing the impacts of vitamin D on HFs is shown in Table 3.

## 5. Stress Fractures

Stress fractures (SFs) are typical injuries in military recruits and committed athletes due to repeated submaximal activity with limited rest. The accumulation of microscopic injuries leads to partial or complete bone fracture, and they mostly occur in the lower limb, including the tibia (23.6%), tarsal navicular (17.6%), metatarsal (16.2) and fibula (15.5%) [82]. It is estimated that 3.3–8.5% of military recruits and 0.5–20% of athletes are affected by SFs [83]. The U.S. Military Services assessed that more than $34,000 USD is lost in training costs for every soldier who suffers from SF [84], and the estimated time to full recovery and unrestricted participation among young athletes who were diagnosed with SF was, on average, 12–13 weeks [85]. There are different risk factors of SFs such as: white ethnicity, female sex, menstrual disturbances, iron deficiency, higher stature, older age and intense physical activity. Among them, an inadequate intake of vitamin D3 is considered a potential and modifiable risk factor [86,87]. What is more, low-serum vitamin D3 is associated with a longer period of recovery from SF [88]. If the women present the female athlete triad (low calorie intake with or without an eating disorder, impaired menstrual cycle and low bone mineral density), the risk of SF increases to 30–50% [89]. The preventive effect of vitamin D3 against SFs was investigated by Lappe et al., who performed a randomized double-blinded study on female Navy recruit volunteers. The subjects were randomized to supplement with 800 IU/d vitamin D3 and 2000 mg calcium or placebo. The study resulted in 20% lower incidences of SF in the experimental group compared to the control group. The limitation of the study was a lack of serum vitamin d3 level measurement [84]. Subsequent results were gained from prospective or retrospective investigations. The study of Sonneville presented that female adolescents (aged 9 to 15 years) whose daily vitamin D3 intake was 663 IU/d, had a 52% lower risk of stress fractures compared with the girls who consumed 107 IU/d. Interestingly, it was found that a higher calcium intake was not protective against SFs [90]. Subsequent research has focused on establishing a threshold for the vitamin D3 serum concentration associated with a reduced SF risk. Since Royal Marine training is perceived as one of the most demanding physically training programs in the world, a study of Davey et al. demonstrated that military recruits whose baseline serum vitamin D3 concentration was below 50 nmol/L (20 ng/mL) had a higher incidence of SFs than cadets with vitamin D3 statuses above this threshold [91]. Likewise, Williams et al. obtained a statistically significant decrease of SF from 7.51% to 1.65% in collegiate athletes who started supplementations of 50,000 IUs of vitamin D3 per week for a period of 8 weeks if their serum vitamin D3 level was <75 nmol/L (<30 ng/mL) [92]. This outcome is in agreement with the results of Ruohola, who found that a serum vitamin D3 concentration < 75 nmol/L (<30 ng/mL) was associated with an increased risk of SF in healthy Finnish military recruits [93]. The higher threshold was reported by Millward et al., who presented that collegiate athletes who improved their vitamin D status to >100 nmol/L (40 ng/mL) had a rate of SF 12% lower than sportsmen who retained a lower serum vitamin D3 status [94]. The same threshold was reported by Miller et al. in their retrospective cohort study among patients who suffered from SF [95]. On the other hand, Grieshober et al. showed no significant association between the serum vitamin D3 concentration and SF history among professional basketball players in the USA. Interestingly, participants with a higher vitamin D3 status had a better chance of being drafted into the National Basketball Association (NBA) [96]. The last conclusion may suggest that vitamin D3 is not only involved in bone health but may also improve the athletic performance, which has already been expanded by some research [25]. According to the mentioned studies, the preventive threshold of the serum D3 level varies across studies and ranges from 50 nmol/L (20 ng/mL) to 100 nmol/L (40 ng/mol) [91,92,93,94,95]. There are a few possible reasons for the discrepancies. First of all, subjects in the abovementioned studies had different training programs, and different types of loads, such as compression, tension and shear forces, were experienced by bones during exercises [97]. Secondly, none of the studies screened individuals for Fok1 and Bsm1 polymorphisms of the vitamin D receptor (VDR), which may be associated with an increased risk of SF [98,99]. Furthermore, women are more likely to suffer from SFs [87], and there were different ratios of men to women in each study.

Finally, vitamin D3 seems to present a preventive effect against SF, but more randomized research is required to establish that property of vitamin D3. However, the measurement of vitamin D3 in the blood and supplementation of vitamin D3 as a routine part of preparation among athletes and military recruits may be beneficial and should be considered. The calcium protective effect against stress fractures and its combination with vitamin D3 remains unclear, and further studies are needed.

A summary of the studies describing the impact of vitamin D on SFs is shown in Table 4.

## 6. Pediatric Fractures

Pediatric fractures (PFs) occur in patients younger than the age of 18, and they account for the most common kind of accidental injury in this group. The peak incidence differs across countries. In the USA, the peak incidence occurs in 10–14 years old [100,101], whereas, in China, in 6–12 years old [102]. The sex ratio (males to females) of PFs is 1:5 [100,103]. PFs are most frequently caused by falls, and the upper limb is the most common affected part of the body [102,103,104]. The prevalence of childhood fractures increases, and it is estimated that half of the pediatric population will suffer from fractures of at least one bone [105]. Interestingly, the frequency of PFs temporarily decreased 2.5 fold during the COVID-19 pandemic possibly due to the reduced use of the playground and participation in organized sports [106]. Apart from taking part in recreation activities and extreme sports, there are several risk factors for PFs, including genetics, obesity, eating disorders, poor nutrition and excessive consumption of sugary drinks [105,107]. A low vitamin D3 concentration as the next risk factor for PF is being discussed widely. Ryan et al. found in African American children that a vitamin D3 deficiency (<50 nmol/L/20 ng/mL) was associated with lower bone mineral density and increased odds of low-energy forearm fractures [108]. Hosseinzadeh et al. investigated pediatric patients with forearm fractures and presented that children who required surgical management were more likely to be vitamin D3-deficient compared to the nonsurgical group (50% versus 17%, respectively). However, children in the surgical group were older and had greater BMIs, which could have influenced the severity of the injuries due to the higher force applied to bones [109]. The association between the serum vitamin D3 level and necessity of a surgical procedure in PF was also shown by Minkowitz et al. They concluded that the serum vitamin D3 level <100 nmol/(40 ng/mL) was correlated with an increased risk for the need for surgical correction. However, a relationship between the vitamin D3 status and fracture occurrence was not found [110]. Interestingly, a case study revealed that the supplementation of vitamin D and calcium in a 9-year-old boy led to bigger endosteal and periosteal callus formation compared to poor callus formation when the same patient suffered from hypovitaminosis D [111]. However, the literature shows contradictory data on the vitamin D3 status and the risk of pediatric fractures. In recent years, a lot of studies have shown no statistically significant differences in the serum vitamin D3 level between children with fractures compared to the control group of pediatric patients without fractures [112,113,114,115,116].

The summary of the studies describing the impact of vitamin D on PFs is shown in Table 5.

## 7. Conclusions

According to this review, more randomized clinical studies must be performed to establish influence of vitamin D on risk of particular types of fractures. One of the main advantages of vitamin D supplementation is its low cost intervention, ease of implementation in daily life and its impact on many aspects of the human body, not only skeletal system. Confirmation of vitamin D effectiveness in the prevention of fractures would be another argument encouraging its supplementation in the population regardless of the amount of sun exposure. Awareness of reducing the risk fractures by only administration of vitamin D could be an effective strategy in the elderly population which is at high risk of osteoporotic fractures. The strategy of reducing SFs among athletes and military recruits by the measurement of vitamin D in the blood and its supplementation may lead to better performance and cost saving. The evaluation of another aspects of vitamin D, especially its impact on recuperation after injuries and hip surgeries is worth of expanding in future studies. Every taken step in reducing early postoperative complications and shortening hospitalization time may improve life quality and decrease mortality rate.

## Figures and Tables

**Table 1 nutrients-15-00103-t001:** Vitamin D supplementation and vertebral fractures—single studies.

Author	Research Design	Study Group	Vitamin D Dose	Serum Vitamin D3 Level	Results
Nakamura et al., 2011 [50]	Cohort	*N* = 773 (773 W) Japanese > 69 years	Averagely 240 IU	Mean 60 nmol/L (24 ng/mL)	Vitamin D3 serum level > 71 nmol/L (28.5 ng/mL) was associated with a reduced risk of VF
Maier et al., 2015 [51]	Retrospective case-control study	Fracture group *N* = 246 Germans (105 M and 141 W) mean age = 69 Control group *N* = 392 Germans (219 M and 173 W) mean age = 63	-	Fracture group 49.1 nmol/L (19.67 ng/mL) Control group 62.6 nmol/L (25.08 ng/mL)	Significant difference in vitamin D3 levels between fracture group and control group; *p* = 0.036
Zhang et al., 2019 [52]	Retrospective case-control study	Fracture group *N* = 534 (108 M and 426 W) mean age = 68.05 Control group *N* = 569 (135 M and 434 W) mean age = 66.84	-	Fracture group 54.53 nmol/L (21.85 ng/mL) Control group 64.56 nmol/L (25.87 ng/mL)	The mean serum vitamin D3 level 29.67 ± 6.18 nmol/L (11.89 ± 2.48 ng/mL) was associated with the higher risk of VF (almost twofold) and more severe fractures. The range of serum vitamin D3 level from 60.91 nmol/L (24.40 ng/mL) to 103.3 nmol/L (41.39 ng/mL) was associated with the lower risk of VF
Hernández et al., 2013 [53]	Cohort	*N* = 820 (820 W) normocalcemic postmenopausal women with high parathyroid hormone levels	-	-	Serum vitamin D3 above 63.6 nmol/L (25.5 ng/mL) decreased bone turnover stimulated by high level of parathormone (>58 pg/mL) and reduced VF prevalence
Atteritano et al., 2017 [54]	Case-control study	*N* = 92 (66 M and 26 W) hemodialysis patients Mean age = 67.10 *N* = 100 controls Mean age = 64.85	Averagely 210 mg/day Averagely 220 mg/day	Subjects with VF 41.43 nmol/L (16.6 ng/mL) Subjects with no fracture 69.39 nmol/L (27.8 ng/mL)	Significant difference in vitamin D3 levels between subjects who suffered from VF and who did not *p* = 0.016
Zafeiris et al., 2012 [55]	Prospective clinical study	*N* = 40 (40 W) postmenopausal women after kyphoplasty Mean age = 70.6	-	Subjects with VF 35.91 nmol/L (14.39 ng/mL) Subjects with no fracture 56.41 nmol/L (22.6 ng/mL)	Significant difference in vitamin D3 levels between subjects who suffered from VF and who did not *p* = 0.001

W-women; M-men; VF-vertebral fracture.

**Table 2 nutrients-15-00103-t002:** Vitamin D supplementation and vertebral fractures—meta-analyses.

Author	Number of Included Studies	Study Group	The Average Age	Vitamin D Dose	Effect Size
Papadimitropoul et al., 2002 [57]	8 RCT placebo vs. vitamin D +/− calcium	*N* = 1130 postmenopausal women	Older than 45 years	>400 IU/day	RR 0.63; CI 0.45–0.88; *p* < 0.01 Significant difference
Jackson et al., 2007 [58]	3 RCT placebo vs. vitamin D +/− calcium	*N* = 1002 postmenopausal women	Older than 60 years	300–800 IU/day	RR 1.22; CI 0.64–2.31 No significant difference
The DIPART group, 2010 [59]	4 RCT Vitamin D + calcium vs. placebo	*N* = 54,493	Mean age = 82	400–800 IU/day	Hazard ratio 0.85 No significant difference
	2 RCT Vitamin D vs. placebo	*N* = 12,880	Mean age = 72.9	800 IU/day	Hazard ratio 1.12 No significant difference
Avenell et al., 2014 [60]	6 RCT Vitamin D vs. placebo	*N* = 11,396	-	-	RR 1.03; CI 0.76–1.39 No significant difference
	2 RCT Vitamin D + calcium vs. calcium	*N* = 2681	-	-	RR 0.14; CI 0.01–2.77 No significant difference
	3 RCT Vitamin D vs. calcium	*N* = 2976	-	-	RR 2.21; CI 1.08–4.53; Vitamin D was less effective than calcium
	4 RCT Vitamin D + calcium vs. placebo	*N* = 42,185	-	-	RR 0.89; CI 0.74–1.09 No significant difference

RCT—randomized controlled trial.

**Table 3 nutrients-15-00103-t003:** Vitamin D supplementation and hip fractures.

Author	Research Design	Study Group	Vitamin D Dose	Serum Vitamin D3 Level	Results
Lilliu et al., 2003 [64]	Randomized controlled trial	Treatment group *N* = 1176 (1176 W) Placebo group *N* = 1127 (1127 W)	800 IU/day	-	25% fewer cases of HFs (138 vs. 184, *p* < 0.02) in the experimental group compared with the placebo group
Rossini et al., 2004 [73]	Quasi-experimental, prospective intervention study	First year *N* = 23,325 (23,325 W) Second year *N* = 24,747 (24,747 W)	Single dose of 400 000 IU/year	-	Decreased frequency of HFs by 10% *p* = 0.05 in comparison with the previous two years
Bischoff et al., 2003 [75]	Double-blind randomized controlled trial	Treatment group *N* = 62 (62 W) Mean age = 85.3 years Placebo group *N* = 60 (60 W) Mean age = 85.3 years	800 IU/day	-	Decreased frequency of falls by 49% in the treatment group compared to the placebo group
Stemmle et al., 2019 [76]	Randomized controlled trial	*N* = 173 after hip surgery Mean age = 84 years	800 IU or 2000 IU/day	-	Clinically significant improvement in the function of the lower limbs
Ingstad et al., 2021 [66]	Retrospective cohort study	*N* = 872 patients with HF Mean age = 80.5 years	Single dose of 100,000 IU	-	Patient after a single-dose of vitamin D had less orthopedic complications during 30 days after surgery (*p* = 0.044; OR 0.32; 95% CI 0.11 to 0.97)
Mak et al., 2016 [77]	Double-blind randomized controlled trial	*N* = 218 patient with HF Mean age = 83.9 years	Single dose of 250,000 IU	-	The treatment group decreased amount of falls compared to the placebo group (6.3% vs. 21.1%; *p* = 0.024) and reduced feeling of pain or discomfort (96.4% vs. 88.8% *p* = 0.037)
Smith et al., 2007 [78]	Double-blind randomized controlled trial	*N* = 9440 (4354 M and 5086 W) >75 years old	Single dose of 300,000 IU vitamin D2	-	No significant association between treatment and control groups in HF frequency (*p* = 0.04)
Meyer et al., 2002 [80]	Randomized controlled trial	Treatment group *N* = 569 Mean age = 84.4 years Placebo group *N* = 575 Mean age = 85.0 years	400 IU/day	-	No significant association between treatment and control groups in HF frequency (*p* = 0.66)

M, men; W, women; HF, hip fracture.

**Table 4 nutrients-15-00103-t004:** Vitamin D supplementation and stress fractures.

Author	Research Design	Study Group	Vitamin D Dose	Serum Vitamin D3 Level	Results
Lappe et al., 2008 [84]	Randomized double-blind study	*N* = 5201 (5201 W) Navy recruits Age range = 17–35 years	800 IU/day	-	The group that supplemented vitamin D had 20% lower incidence of SF than the control group (5.3% vs. 6.6%; *p* = 0.0026)
Sonneville et al. 2012 [90]	Prospective cohort study	*N* = 6712 (6712 W) Age range = 9–15 years	Treatment group 663 IU/day Control group 107 IU/day	-	The treatment group had 52% lower risk of SF than control group HR = 0.48; 95% CI, 0.22–1.02; *p* = 0.04 No significant difference between calcium intake and SF
Davey et al. 2016 [91]	Prospective cohort study	*N* = 1082 (1082 M) Royal Marine Age range = 16–32 years	-	Mean 69.2 ± 29.2 nmol/L (27.2 ± 11.7 ng/mL)	Serum vitamin D3 level <50 nmol/L (20 ng/mL) was associated with an increased risk of SF compared to subjects with above this threshold; *p* = 0.042; odds ratio 1.6 (95% confidence interval 1.0–2.6)
Williams et al., 2020 [92]	Prospective cohort study accompanied by a retrospective review for control comparison	*N* = 118 (30 M, 88 W) Mean age = 19.7 years in M Mean age = 19.6 years in W Retrospective control group *N* = 453	50 000 IUs of vitamin D3 per week for a period of 8 weeks in subjects with serum level of vitamin D3 < 75 nmol/L (<30 ng/mL)	Mean 80.37 nmol/L (32.2 ng/mL) in August and mean 79.62 nmol/L (31.9 ng/mL) in February	Decrease of SF from 7.51% to 1.65% (*p* = 0.009) in the treatment group compared to the control group
Ruohola et al., 2006 [93]	Prospective cohort study	*N* = 800 (800 M) Finnish military recruits Mean age = 19.8 years	-	Mean 75.8 nmol/L (30.4 ng/mL)	Serum vitamin D3 level <75 nmol/L (<30 ng/mL) was associated with increased risk of SF OR 3.6 (95% CI: 1.2–11.1)
Millward et al., 2020 [94]	Prospective cohort study	*N* = 802 (497 M and 305 W) collegiate athletes	50,000 IUs or 30,000 IUs of vitamin D3 per week for a period of 8 weeks in subjects with serum level of vitamin D3 <50 nmol/L (20 ng/mL) or <75 nmol/L (<30 ng/mL) respectively	Mean 93.6 nmol/L (37.5 ng/mL) in M Mean 108.6 nmol/L (43.5 ng/mL) in W	Collegiate athletes who improved their vitamin D3 status to >100 nmol/L (40 ng/mL) had the rate of SF 12% lower than sportsmen who remained low serum vitamin D3 status (95% CI, 6–19; *p* < 0.001)
Miller et al., 2016 [95]	Retrospective cohort study	*N* = 124 (42 M and 82 W)	-	Mean 31.14 ± 14.71 nmol/L (12.48 ± 5.89 ng/mL)	Vitamin D3 status < 40 nmol/L (16 ng/mL) was associated with an increased risk of SF
Grieshober et al., 2018 [96]	Descriptive epidemiology study	*N* = 279 (279 M) professional basketball players	-	-	No significant association between serum vitamin D3 concentration and stress fracture

M, men; W, women; SF, stress fracture.

**Table 5 nutrients-15-00103-t005:** Vitamin D supplementation and pediatric fractures.

Author	Research Design	Study Group	Vitamin D Dose	Serum Vitamin D3 Level	Results	
Ryan et al., 2012 [108]	Case-control study	Fracture group *N* = 76 (44 M, 32 F) Control group *N* = 74 (40 M, 34 F) Age range = 5–9 years	-	-	Serum vitamin D3 level < 50 nmol/L (20 ng/mL) was associated with lower bone mineral density and increased odds of low energy forearm fractures (47.1% vs. 40.8%; adjusted odds ratio 3.46)	
Hosseinzadeh et al., 2020 [109]	Prospective cohort study	*N* = 100 (65 M, 35 F) Age range = 3–15 years	-	Mean 68.6 ± 20.7 nmol/L (27.5 ± 8.3 ng/mL)	Children who required surgical management were more likely to be vitamin D3 deficient compared with the nonsurgical group (50% versus 17% respectively); RR of surgical treatment in children with forearm fracture and D3 deficiency was 3.8	
Minkowitz et al., 2017 [110]	Retrospective and prospective cohort study	Fracture group *N* = 369 Control group *N* = 662	-	Fracture group 68.8 nmol/L (27.5 ng/mL) Control group 68.4 nmol/L (27.4 ng/mL)	Serum vitamin D3 level < 100 nmol/L(40 ng/mL) was correlated with increased risk for need for surgical correction. No relationship between vitamin D3 status and PF occurrence.	
Ceroni et al., 2012 [112]	Prospective cohort study	Fracture group N = 100 Mean age = 12.9 years Control group N = 50 Mean age = 12.7 years	-	Fracture group 80.1 nmol/L (32.1 ng/mL) and 76.1 nmol/L (30.5 ng/mL) Control group 81.6 nmol/L (32.7 ng/mL)	No significant association between serum vitamin D3 concentration and PF	
Perez-Rossello et al., 2012 [113]	Prospective cohort study	*N* = 40 (18 M, 22 F) Age range = 8–24 months	-	-	No fracture before or after treatment for vitamin D3 deficiency	
Contreras et al., 2014 [114]	Prospective case-control study	Fracture group *N* = 100 Control group *N* = 100	-	Fracture group 66.6 nmol/L (26.7 ng/mL) Control group 63.6 nmol/L (25.5 ng/mL)	No significant association between serum vitamin D3 concentration and PF *p* = 0.859; odds ratio 0.94)	
Moore et al., 2014 [115]	Prospective case-control study	Fracture group *N* = 58 Mean age = 8.0 years Control group *N* = 58 Mean age = 8.9 years	-	Fracture group 63.2 nmol/L (25.3 ng/mL) Control group 62.5 nmol/L (25.0 ng/mL)	No significant association between serum vitamin D3 concentration and pediatric fracture *p* = 0.86	
Schilling et al., 2011 [116]	Cohort	*N* = 118 Mean age = 203 days	-	-	No significant association between serum vitamin D3 concentration and pediatric fracture *p* = 0.32	

M, male; F, female; PF, pediatric fracture; RR, relative risk.

## Data Availability

Not applicable.

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
