# Peer review of "Vitamin D Supplementation and Its Impact on Different Types of Bone Fractures"

_nutrients, 2022, doi:10.3390/nu15010103_

Round 1
Reviewer 1 Report
Dear Editors and authors,
I am grateful to revise this paper.
This review concerning the impact of vitamin D status on global fracture risk is intriguing and almost complete, providing another important chance to update physicians about the state of the art regarding the challenging and neverending “vitamin D issue”.
Thus, in my opinion, this paper could be suitable for publication after few revisions.
In fact, it could be improved as follows:
· Introduction: it could be appropriate to shorten this section, by synthesizing the concepts about metabolism of vitamin D.
· In my opinion, a specific section regarding “Materials and methods” should be added.
· Discussion: in my opinion, the authors could improve the quality of their manuscript pointing out the importance of a more clinical comprehensive approach regarding the management of fracture risk, taking into account the availability of many algorithms, like FRAX, that could help clinicians to better assess risk, even regardless of vitamin D status (See and cite Lorentzon M. Treating osteoporosis to prevent fractures: current concepts and future developments. J Intern Med. 2019 Apr;285(4):381-394. doi: 10.1111/joim.12873. Epub 2019 Jan 18. PMID: 30657216.; Sagalla N, Colón-Emeric C, Sloane R, Lyles K, Vognsen J, Lee R. FRAX without BMD can be used to risk-stratify Veterans who recently sustained a low trauma non-vertebral/non-hip fracture. Osteoporos Int. 2021 Mar;32(3):467-472. doi: 10.1007/s00198-020-05616-5. Epub 2020 Sep 4. PMID: 32885318; PMCID: PMC7930138.)
· Additionally, they should underline that an adequate approach to vertebral fractures should involve different experts, in view of their possible multidisciplinary consequences (See and cite Capozzi A, Scambia G, Pedicelli A, Evangelista M, Sorge R, Lello S. Clinical management of osteoporotic vertebral fracture treated with percutaneous vertebroplasty. Clin Cases Miner Bone Metab. 2017 May-Aug;14(2):161-166. doi: 10.11138/ccmbm/2017.14.1.161. Epub 2017 Oct 25. PMID: 29263726; PMCID: PMC5726202.; Dumitrescu B, van Helden S, ten Broeke R, Nieuwenhuijzen-Kruseman A, Wyers C, Udrea G, van der Linden S, Geusens P. Evaluation of patients with a recent clinical fracture and osteoporosis, a multidisciplinary approach. BMC Musculoskelet Disord. 2008 Aug 5;9:109. doi: 10.1186/1471-2474-9-109. PMID: 18680609; PMCID: PMC2529301.
· Finally, I suggest reviewing English form.
Sincerely
Author Response
Dear Sir / Madam,
Firstly I would like to thank you for your positive feedback and relevant suggestions. I am sending you the manuscript with the adjustments made.
In fact, it could be improved as follows:
- Introduction: it could be appropriate to shorten this section, by synthesizing the concepts about metabolism of vitamin D.
We have shorten the introduction by dividing it into 2 parts to make it more transparent.
- In my opinion, a specific section regarding “Materials and methods” should be added.
We agree with your suggestion, however the articles chosen by us were picked basing on our clinical experience and knowledge.
- Discussion: in my opinion, the authors could improve the quality of their manuscript pointing out the importance of a more clinical comprehensive approach regarding the management of fracture risk, taking into account the availability of many algorithms, like FRAX, that could help clinicians to better assess risk, even regardless of vitamin D status (See and cite Lorentzon M. Treating osteoporosis to prevent fractures: current concepts and future developments. J Intern Med. 2019 Apr;285(4):381-394. doi: 10.1111/joim.12873. Epub 2019 Jan 18. PMID: 30657216.; Sagalla N, Colón-Emeric C, Sloane R, Lyles K, Vognsen J, Lee R. FRAX without BMD can be used to risk-stratify Veterans who recently sustained a low trauma non-vertebral/non-hip fracture. Osteoporos Int. 2021 Mar;32(3):467-472. doi: 10.1007/s00198-020-05616-5. Epub 2020 Sep 4. PMID: 32885318; PMCID: PMC7930138.)
I have expanded references and have mentioned available fracture risk assessment tools.
- Additionally, they should underline that an adequate approach to vertebral fractures should involve different experts, in view of their possible multidisciplinary consequences (See and cite Capozzi A, Scambia G, Pedicelli A, Evangelista M, Sorge R, Lello S. Clinical management of osteoporotic vertebral fracture treated with percutaneous vertebroplasty. Clin Cases Miner Bone Metab. 2017 May-Aug;14(2):161-166. doi: 10.11138/ccmbm/2017.14.1.161. Epub 2017 Oct 25. PMID: 29263726; PMCID: PMC5726202.; Dumitrescu B, van Helden S, ten Broeke R, Nieuwenhuijzen-Kruseman A, Wyers C, Udrea G, van der Linden S, Geusens P. Evaluation of patients with a recent clinical fracture and osteoporosis, a multidisciplinary approach. BMC Musculoskelet Disord. 2008 Aug 5;9:109. doi: 10.1186/1471-2474-9-109. PMID: 18680609; PMCID: PMC2529301.
We have highlighted the necessity of multidisciplinary approach to diagnose underlying medical conditions
- Finally, I suggest reviewing English form.
We have corrected minor English mistakes which we noted.
Sincerely
Reviewer 2 Report
This study with the title of vitamin D supplementation and type of bone fractures is very innovative. So regarding of minor English language edition, it will be great for publication.
Author Response
Dear Sir / Madam,
Thank you for your positive feedback and bringing these mistakes to our attention. We have corrected minor English mistakes which we noted.
Sincerely
Reviewer 3 Report
Well written paper. Congratulation.
The Main question was clear. I would be more interested in the different bone fractures in the pediatric population as well. As I mentioned my opinion in the review.
This article is original, but previous studies highlight the importance of calcium metabolism. I am interested in any metabolism pathways that play on bone stabilization or metabolism.
It does not add as much to the subject area as compared to other publications
The Authors highlight the fracture (they wrote some examples vertebral, hip so on, and finally conclude pediatric fracture. It would have been better to compare adult and childhood fractures.
The conclusions consistent with the evidence and arguments presented and they address the main question posed.
I suggested some more add to the reference about PACAP effect of the bone metabolism.
Author Response
Well written paper.
Congratulation
Dear Sir / Madam,
Thank you for your positive feedback.
Sincerely